# Exploring the knowledge and attitudes of Cameroonian medical students towards global surgery: A web-based survey

Ulrick S. Kanmounye[1,2]*, Aimé N. Mbonda[3,4º], Dylan Djiofack[2,3º], Leonid Daya[3,5‡], Ornella F. Pokam[3,6‡], Nathalie C. Ghomsi[2,3,7]

**1** Research Team, International Student Surgical Network—International Team, Boston, Massachusetts, United States of America, **2** Research Department, Association of Future African Neurosurgeons, Yaounde, Centre, Cameroon, **3** Research Department, International Student Surgical Network—National Working Group, Yaounde, Centre, Cameroon, **4** Department of Public Health, Faculty of Medicine and Biomedical Sciences, University of Yaounde I, Yaounde, Centre, Cameroon, **5** Department of Anesthesiology, Faculty of Medicine and Biomedical Sciences, University of Yaounde I, Yaounde, Centre, Cameroon, **6** Faculty of Health Sciences, University of Buea, Buea, South West, Cameroon, **7** Department of Neurosurgery, Felix Houphouet Boigny University, Abidjan, Côte d'Ivoire

º These authors contributed equally to this work.
‡ These authors also contributed equally to this work.
* ulricksidney@gmail.com

**Data Availability Statement:** The underlying data are available in the Supporting Information files.

**Funding:** The author(s) received no specific funding for this work.

## Abstract

### Introduction

Global surgery is a growing field studying the determinants of safe and affordable surgical care and advocating to gain the global health community's attention. In Cameroon, little is known about the level of knowledge and attitudes of students. Our survey aimed to describe the knowledge and attitudes of Cameroonian medical students towards global surgery.

### Materials and methods

We performed an anonymous online survey of final-year Cameroonian medical students. Mann-Whitney U test and Spearman correlation analysis were used for bivariate analysis, and the alpha value was set at 0.05. Odds ratios and their 95% confidence intervals were calculated.

### Results

204 respondents with a mean age of 24.7 years (±2.0) participated in this study. 58.3% were male, 41.6% had previously heard or read about global surgery, 36.3% had taken part in a global surgery study, and 10.8% had attended a global surgery event. Mercy Ships was well known (46.5%), and most students believed that surgical interventions were more costly than medical treatments (75.0%). The mean score of the global surgery evaluation was 47.4% (±29.6%), and being able to recognize more global surgery organizations was correlated with having assumed multiple roles during global surgery studies (p = 0.008) and identifying more global surgery indicators (p = 0.04). Workforce, infrastructure, and funding were highlighted as the top priorities for the development of global surgery in Cameroon.

**Competing interests:** The authors have declared that no competing interests exist.

## Conclusion

Medical students are conscious of the importance of surgical care. They lack the opportunities to nurture their interest and should be taught global surgery concepts and skills.

## Introduction

Two-thirds of the world's population lacks access to safe, timely, and affordable surgical, obstetric, and anesthetic (SOA) care. [1] SOA public health interventions have traditionally been viewed as complicated and expensive, [2], and this has led to the relegation of SOA amenable diseases on national and international public health agendas. Global surgery is "an area for study, research, practice, and advocacy that seeks to improve health outcomes and to achieve health equity for all people requiring surgical care, with a special emphasis on underserved populations and populations in crisis." [3] Eleven years ago, global surgery was referred to as the "neglected stepchild of global health." [4] In 2015 however, the Lancet Commission on Global Surgery published a seminal report that emphasized the burden of SOA amenable diseases, defined six global indicators and corresponding targets to be attained by 2030 and advocated for health systems strengthening through national surgical, obstetrics and anesthesia plans (NSOAPs). [5] During the same year, the World Health Assembly passed resolution WHA 68.15, which emphasized the importance of SOA care in achieving universal health coverage. [6] As we strive to achieve universal health coverage, we must remember that "no country can achieve universal health coverage unless its people have access to safe, timely and affordable [SOA] services." [7]

Global surgery transcends the operating room and the perioperative period spatially and temporally and addresses the causes of the three delays of access to SOA care: delay in seeking, reaching, and accessing safe, timely, and affordable SOA care. NSOAPs "provide a vision, along with costed- and time-bound targets, of how actors within the SOA system will work together to systematically improve the SOA health system." [8] Practically, this translates to eight steps: securing support from relevant government institutions; evaluation of the current SOA system; involvement of SOA system actors and priority setting; design and approval of a plan; monitoring and evaluation; budgeting; improved governance; and implementation. [9] Locally driven research, advocacy at high-level meetings and the ministerial level, and education aimed at health systems strengthening in underserved populations are indispensable to the development of effective NSOAPs.

Cameroon is a Central African country with a population of 25 million. [10] Most Cameroonians live within 2-hours of a public, a private for-profit, a private not-for-profit, or a faith-based healthcare facility offering essential SOA care. [11] This geographical proximity is, in reality, limited by the lack of financial risk protection as 63% of total SOA care expenditures are paid out-of-pocket by patients and their families. [12] Surgical safety is a concern in Cameroon, given the 7-day postoperative complication rate is at 13.1%, and the postoperative mortality rate is at 2%. [13] While these numbers are similar to those of other African countries, they remain high when compared to those of high-income countries. During the last human resources for health census in 2011, Cameroon had 153 surgeons, 140 obstetricians, and 22 anesthesiologists [14]. Public, faith-based, and private secular organizations offer surgical care in Cameroon. Faith-based organizations offer surgical care for free or at reduced costs, mostly in rural areas and to low-income families in urban areas. Since medical students form the national potential in workforce capacity development, their attitudes towards global surgery

are of paramount importance. Despite the growing interest in global surgery in the country, no data is currently available on the knowledge and attitudes of medical students towards global surgery.

## Materials and methods

The institutional review board of Université des Montagnes authorized this study. We carried out a literature review to inform our questionnaire on landmark global surgery findings [1,2,5,13,15,16], and we submitted the questions to three experts who established the face validity of the survey. The online survey was anonymous, administered in official languages (French and English), and hosted on Google Forms (Alphabet Inc., California, USA). We then piloted our survey among 20 Cameroonian medical students. Responses to the pilot survey were excluded, and all issues raised during the pilot were addressed. We distributed the survey link to class representatives of our target population (final year medical students in Cameroon). The class representatives then went on to distribute the survey links in the official class WhatsApp groups (Facebook Inc., California, USA). WhatsApp is the primary means of formal communication among medical students, so all students join the official class group. We collected the total number of students in each group to calculate the participation rate, link dissemination was confirmed by screenshots, and reminder messages were sent every week in the class groups to ensure a reasonable participation rate.

We collected sociodemographic information, assessed knowledge, and perceptions of our respondents about global surgery. Survey questions were a mix of open, close, multiple-choice, and Likert scale questions. Data on negatively phrased questions were reversed coded to ensure consistency with positively phrased questions that were similar. The data collected were coded and tidied on Excel 2016 (Microsoft Corp., Washington, USA). We analyzed bivariate data using the Mann-Whitney U test and Spearman correlation on SPSS version 24 (IBM, New York, USA). The threshold of significance was set at 0.05. Data collection complied with the terms and conditions of Google Forms.

## Results

We received 204 responses (participation rate: 84.3%, n = 242) from final year medical students enrolled at all of the six Cameroonian medical schools. The students resided in seven of the ten Cameroonian regions (Fig 1), they had a mean age of 24.7 years (±2.0), and 119 (58.3%) were male. 91 students (41.6%) had previously heard or read about global surgery, 74 (36.3%) had previously participated in a global surgery study, and 22 (10.8%) had attended a global surgery event. Only 15 students (7.4%) had both previously participated in a global surgery study and attended a global surgery event. Table 1 summarizes the students' global surgery experiences by sex.

46 students (22.5%) defined "global surgery" correctly. We considered a definition to be correct if it used the following set of terms or their synonyms: "surgery," "everyone," "everywhere," and "access." Global surgery was most often mistaken to mean "general surgery" or was thought to be "a multidisciplinary surgical team." 32 respondents (15.7%) could not identify or cite a global surgery organization, whereas 106 (52.0%) could only identify one organization. The most commonly cited organizations were Mercy Ships (46.5%), InciSioN (21.3%), and the Lancet Commission on Global Surgery (8.9%). 162 students (79.4%) recognized a single global surgery indicator. Access to timely essential surgery was the most widely known global surgery indicator (39.3%). Other popular indicators included perioperative mortality rate (25.5%), specialist surgical workforce density (16.1%), and risk of impoverishing expenditure (7.1%). The risk of catastrophic expenditure (6.4%) and the surgical volume (5.6%) were the least popular of all global surgery indicators. Four students (2.0%) had read or heard about

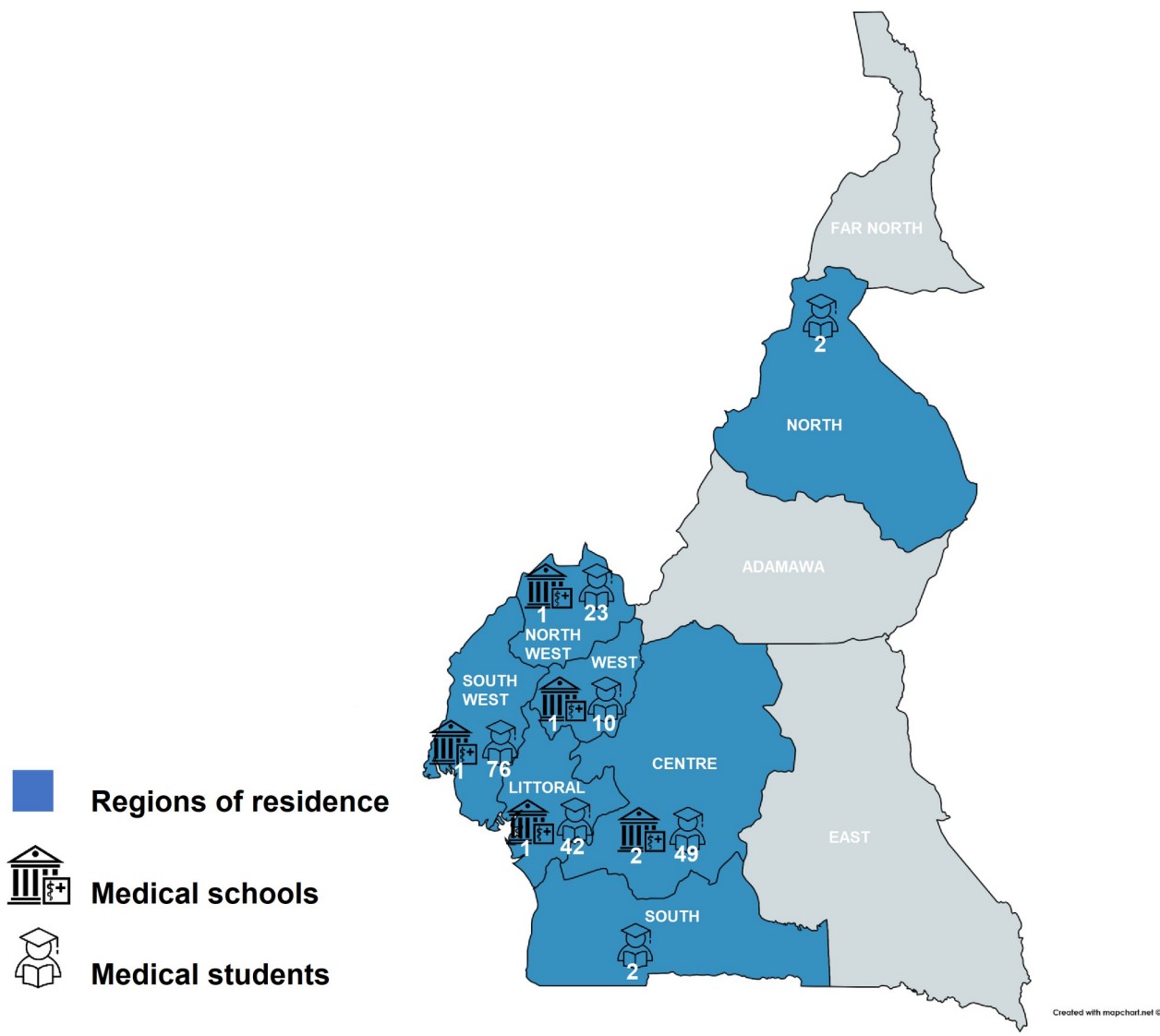

**Fig 1. Medical school locations and permanent residencies of medical students.** Regions of permanent residence are colored in blue. Buildings represent medical schools, and icon heads donning graduate caps represent medical students. The number of medical students and medical schools is placed below the corresponding icons.

national surgical, obstetric, and anesthesia plans, while only 3 students (1.5%) knew about the disease control priorities 3 essential surgeries.

Students were asked questions that were based on the key findings of landmark global surgery papers. The mean score for this quiz was 47.4% (±29.6%), with 108 students (52.9%) performing below average. Unsurprisingly, having heard or read about global surgery (p = 0.047) was significantly associated with a better performance at the quiz, unlike participation in a global surgery study (p = 0.68) and attendance at a global surgery event (p = 0.34). Recognition of global surgery organizations was correlated with having assumed more roles (collaborator, data collector, or investigator) in global surgery studies (p = 0.008) and recognition of global surgery indicators (p = 0.04).

**Table 1. Global surgery experiences of cameroonian final year medical students.**

| Characteristic | Female (%) | Male (%) | OR (95% CI) [a] |
|---|---|---|---|
| Heard or read about global surgery (via) | 38 (18.63) | 53 (25.98) | 0.99 (0.57–1.73) |
| Acquaintance | 9 | 20 | |
| Hospital | 10 | 9 | |
| School | 18 | 18 | |
| Social media | 13 | 42 | |
| Television | 5 | 2 | |
| Other | 4 | 3 | |
| Number of media through which students learned about global surgery | | | |
| 1 | 31 (15.19) | 33 (16.17) | |
| 2 | 4 (1.96) | 5 (2.45) | |
| 3 | 2 (0.98) | 8 (3.92) | |
| 4 | 2 (0.98) | 4 (1.96) | |
| 5 | 1 (0.49) | 2 (0.98) | |
| Attendance at a global surgery event | 7 (3.43) | 15 (7.35) | 1.61 (0.63–4.13) |
| In-person in Cameroon | 5 | 9 | |
| In-person abroad | 0 | 1 | |
| Online | 2 | 5 | |
| Participation in global surgery research | 27 (13.24) | 47 (23.04) | 1.40 (0.78–2.52) |
| Collaborator | 5 | 11 | |
| Data collector | 8 | 13 | |
| Investigator | 5 | 7 | |
| Respondent | 12 | 17 | |
| Number of roles assumed by the students | | | |
| 1 | 25 (12.25) | 36 (17.65) | |
| 2 | 0 | 6 (2.94) | |
| 3 | 1 (0.49) | 2 (0.98) | |

[a] Odds ratio for sex (Female/Male)

Most students (62.7%) believed surgically amenable diseases were a significant public health problem in Cameroon; however, 153 (75.0%) felt that surgical interventions cost more than medical treatments. 157 (77.0%) supported task-shifting as a solution to lessen the burden of surgically amenable diseases. The students identified workforce, investment in tertiary level facilities, and funding as priorities for the attainment of universal access to surgical, obstetric, and anesthetic care in Cameroon (Fig 2).

## Discussion

The majority of final year Cameroonian medical students have never heard or read about global surgery. They learned about global surgery primarily on social media, and few have had the opportunity to participate in global surgery research or to attend global surgery events. The fact that social media ranked ahead of medical school and teaching hospital lectures reinforces the notion that social media is a tool for research, education, and advocacy dissemination in global surgery. [17] We recommend a change in the undergraduate curriculum to include at least one lecture on global surgery. Mercy Ships was the most widely known global surgery institution among medical students. The popularity of Mercy Ships can be attributed to its recent year-long surgical mission to Cameroon. [18]

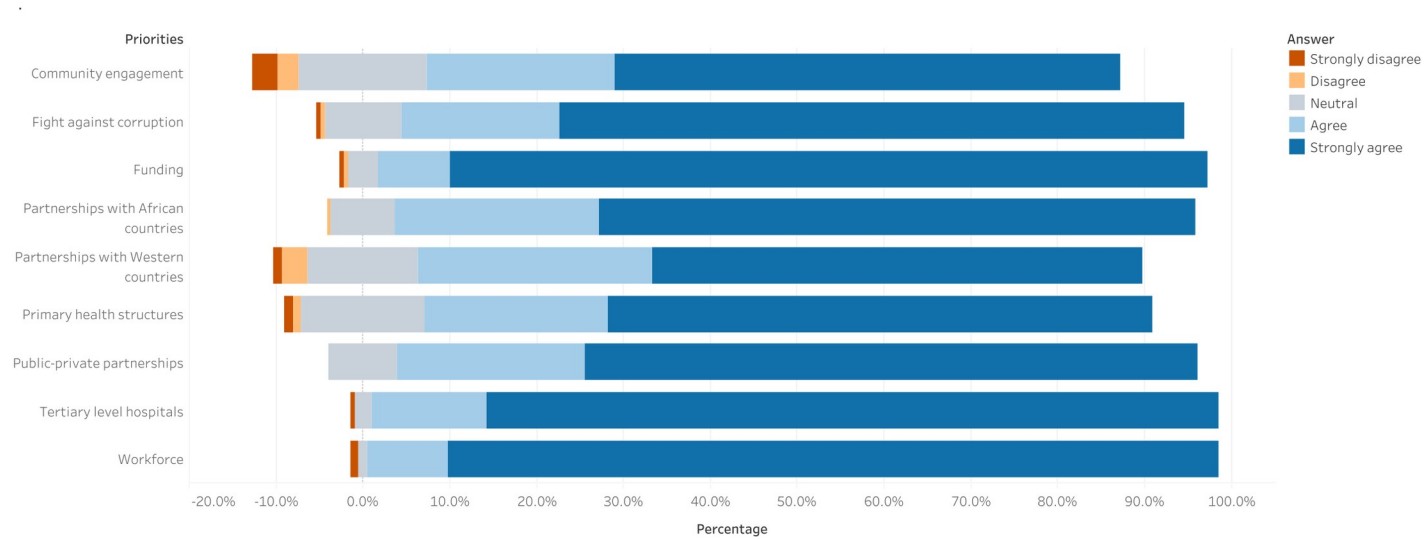

**Fig 2. Global surgery priorities in Cameroon according to final year medical students.**

Generally, respondents were not comfortable dealing with global surgery concepts and literature. Unsurprisingly, students who had more global surgery research experience performed better at the quiz. The subpar performance of the respondents could be explained by the lack of global surgery research and education opportunities. Exposure to global surgery has been shown to raise the interest of future global surgeons. [19] Despite the popularity of global surgery, opportunities are scarce. Students at American and Canadian schools report having limited global surgery opportunities. [20,21] Among the few global surgery opportunities in the global surgery course developed by Lund University, Harvard Medical School, and the University of Zimbabwe. [22] This course is run for five weeks, and it combines theory and practice in a low resource setting. Another opportunity is the research associate fellowships offered by the Program in Global Surgery and Social Change, Harvard Medical School. This position goes to 5–8 senior medical students from all over the world each year. [23] Global surgery conferences equally represent an opportunity for students to learn. Students gain valuable skills at global surgery academic events. [21,24] Some global surgery conferences include the COSECSA annual meeting, the Consortium of Universities in Global Health, Bethune round table, InciSioN Global Surgery Symposium, and Global Surgery Student Alliance symposium.

Students are already contributing to global surgery research and education. The British student research collaborative, STARSurg, is currently running global pediatric surgery studies in Tanzania. [25] In Rwanda, students assisted their government during the NSOAP design, collecting data all around the country. Pleased with the contribution of medical students, the government went on to integrate strategies aimed at increasing global surgery involvement among medical students in the NSOAP. [26]

Surgically amenable diseases affect people of all ages, genders, and nationalities. However, most of the 5 billion people who have limited access to safe, timely, and affordable SOA care are either citizens of low- and middle-income countries or residents of underserved regions in high-income countries. [1] Medical students from these regions can contribute to the attainment of universal access to surgical care if they are given the skills and opportunities. Social media play a critical role in the dissemination of global surgery research, advocacy, and education among medical students.

We acknowledge the following limitations in our study. First, we contacted students via the class WhatsApp groups. This mode of dissemination might have led to undercoverage bias, or nonresponse bias of our study population given as individuals from lower-income groups might not be able to afford data plans. We reduced nonresponse bias by sending weekly reminder messages, and we obtained students' phone numbers and the total number of enrolled students from class delegates. Our participation rate suggests that the risks of these biases were minimal in our survey. Other limitations include variations in understanding the questions, the limitations of areas of questioning encompassed by the survey, and the unproven generalizability of our findings to other LMICs.

## Conclusion

The majority of final year medical students do not understand or know what global surgery is. Few students have had the opportunity to participate in global surgery research or events, but students are aware of the health and social impacts of surgically amenable diseases. When they are given access to resources, students contribute to research, education, and advocacy in the field and are more likely to pursue a global surgery career. The emphasis, therefore, should be laid on increasing the involvement of medical students in global surgical education and activities.

## Supporting information

**S1 Dataset.**
(XLSX)

**S1 File.**
(PDF)

## Acknowledgments

We wish to thank Dominique Vervoort for his help in proofreading this manuscript. We wish to acknowledge Anudari Zorigtbaatar, Nabeel Muhammad, and Desmond Jumbam for their help with the face validity of the survey.

## Author Contributions

**Conceptualization:** Ulrick S. Kanmounye, Aimé N. Mbonda, Leonid Daya, Ornella F. Pokam.

**Data curation:** Ulrick S. Kanmounye.

**Formal analysis:** Ulrick S. Kanmounye.

**Investigation:** Ulrick S. Kanmounye, Aimé N. Mbonda, Dylan Djiofack, Ornella F. Pokam.

**Methodology:** Ulrick S. Kanmounye.

**Project administration:** Ulrick S. Kanmounye.

**Resources:** Ulrick S. Kanmounye.

**Software:** Ulrick S. Kanmounye.

**Supervision:** Ulrick S. Kanmounye, Aimé N. Mbonda.

**Validation:** Ulrick S. Kanmounye, Dylan Djiofack, Leonid Daya, Nathalie C. Ghomsi.

**Visualization:** Ulrick S. Kanmounye.

**Writing – original draft:** Ulrick S. Kanmounye, Dylan Djiofack, Leonid Daya, Ornella F. Pokam, Nathalie C. Ghomsi.

**Writing – review & editing:** Ulrick S. Kanmounye, Dylan Djiofack, Leonid Daya, Nathalie C. Ghomsi.

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
