## [Decision Letter · Decision Letter 0]

9 Jan 2020

PONE-D-19-34339

Knowledge and Attitudes of Medical Students and Healthcare Workers Towards Global Surgery in Cameroon

PLOS ONE

Dear Dr. Kanmounye,

Thank you for submitting your manuscript to PLOS ONE. After careful consideration, we feel that it has merit but does not fully meet PLOS ONE’s publication criteria as it currently stands. Therefore, we invite you to submit a revised version of the manuscript that addresses the points raised during the review process.

We would appreciate receiving your revised manuscript by Feb 23 2020 11:59PM. To enhance the reproducibility of your results, we recommend that if applicable you deposit your laboratory protocols in protocols.io, where a protocol can be assigned its own identifier (DOI) such that it can be cited independently in the future. For instructions see: http://journals.plos.org/plosone/s/submission-guidelines#loc-laboratory-protocols

We look forward to receiving your revised manuscript.

Kind regards,

James G. Wright

Academic Editor

PLOS ONE

Additional Editor Comments:

There are significant issues identified by the reviewers. I want to highlight two

1. This study would not be appropriate for publication in PLOS ONE unless the application of the findings would be wider than Cameroon.

2. If the authors would want to have the study published in this journal, I would suggest redoing the questionnaire, and showing clearly how questions were asked. They could then administer it to a homogenous group such as all final year medical students in the country, and find a way to achieve a high response rate (preferably 80% or above).

2. Please include additional information regarding the survey or questionnaire used in the study and ensure that you have provided sufficient details that others could replicate the analyses.

For instance, if you developed a questionnaire as part of this study and it is not under a copyright more restrictive than CC-BY, please include a copy, in both the original language and English, as Supporting Information. 

If the original language is written in non-Latin characters, for example Amharic, Chinese, or Korean, please use a file format that ensures these characters are visible.

3. Please state whether you validated the questionnaire prior to testing on study participants.

Please provide details regarding the validation group within the methods section.

4. In your Methods section, please include additional information about your dataset and ensure that you have included a statement specifying whether the collection method complied with the terms and conditions for the website.

Reviewers' comments:

Reviewer's Responses to Questions

**Comments to the Author**

1. Is the manuscript technically sound, and do the data support the conclusions?

Reviewer #1: Yes

Reviewer #2: No

2. Has the statistical analysis been performed appropriately and rigorously? 

Reviewer #1: No

Reviewer #2: N/A

3. Have the authors made all data underlying the findings in their manuscript fully available?

Reviewer #1: Yes

Reviewer #2: No

4. Is the manuscript presented in an intelligible fashion and written in standard English?

Reviewer #1: Yes

Reviewer #2: Yes

5. Review Comments to the Author

Reviewer #1: The paper reads well but is primarily a discussion on global surgery. There is clearly a need to develop surgical services in Cameroon, but does the workforce need to advocate global surgery or to focus on developing surgical services in Cameroon itself? The latter is likely more important. There should be details included of the plans for Cameroon to develop their surgical services and whether the government is guided by global health targets.

The contribution of Mercy Ships is well made through the lead in training and providing a vital service for the local community.

I believe this paper is worthy of publication but more as a discussion of the need for countries (like Cameroon) to work towards achieving Global Surgery targets. The questionnaire is more of a side discussion.

Statistics are included in the paper but do not add to the analysis and should be removed.

Reviewer #2: I am interested that the lead researcher appears to be based in DR Congo and ethics was approved there, although the study was conducted in Cameroon. This seems to conflict with line 117 where the manuscript states that local ethical approval was obtained.

The introduction is nicely constructed and builds to the purpose of this study.

It would be very helpful to have some workforce data for Cameroon. Eg what percentage of doctors registered in Cameroon work in anaesthetic, obstetric or surgical disciplines?

Line 92 I am unsure what the authors mean when they say that global surgery in Cameroon is ‘led’ by InciSioN? Do they mean that InciSioN is one of the leading pressure groups promoting global surgery?

Line 101-102: I feel that the authors could emphasize the wider potentials for the study. This study would not be appropriate for publication in PLOS ONE unless the application of the findings would be wider than Cameroon.

In Methods the authors first describe how they designed their survey line107. This section is brief and the reader only really knows which literature was read.

The questionnaire itself needs to be seen so that readers can see how questions were expressed, and could repeat the study in their own country or at a different time.

The authors then describe how they found their participants for the study. Their description is not very clear – either in terms of how they selected or accessed their target audience. They say that they used a convenience non-probability sampling method but do not reference this or explain how it worked , line 112.

They do not state how many people received the questionnaire, and hence the response rate is not evident.

It is very hard to know how representative their sample is likely to be.

In line 112,113 I am unsure why they chose medical Students and nurses and doctors including surgeons. It seemed to be anyone who was willing to complete the survey. The study would have more clarity if it was a single target audience. It is hard to imagine the respondents did not self-select as those interested in the topic. The groups sampled cross professions and training/trained demographics. For the medical students we are not told in which year of study they are currently. Clearly a first year medical student would have less chance of understanding global surgery than a final year student. We are not told if there is teaching on this subject in the medical student curriculum, and if so in which year of study it is taught?

The forms were anonymous but the authors say that they omitted members of the InciSioN group – how did they do this?

Results

Line 121 there were only 45 respondents. We are not informed how many medical students are present in Cameroon, or indeed health care workers.

Line 129: do the authors mean most popular (well thought of) or popular (well known).

In the table:

Health professionals presumably means 18 doctors and one nurse?

The question ‘do you know about ‘global surgery’ appears confusing, and might be better expressed ‘have you previously heard of the term ‘global surgery’?’

For all the summarised section results it is unclear whether or not these are based on open or closed questions. For example were the organisations named and the respondents stated whether or not they had heard of them, or were they free text responses conceived by respondents?

Line 144 some results are given in the text but not in the table whilst others are in the table and repeated in the text. This needs to be consistent.

Having not seen the questionnaire I am unsure whether any questions explored the place of trauma and injury in surgically treatable disease. This is particularly relevant to the perception of surgically treatable disease as a public health issue.

Discussion

The discussion is quite long but rather mono-focussed.

There is no comment on similar literature.

There is some discussion regarding why respondents responded as they did.

Much of the focus seems to bear on proposed solutions to spread the word regarding global surgery, and a bit of publicity for InciSioN Cameroon. It would be helpful if the authors explored the issue of current knowledge and understanding of global surgery without such particular emphasis on InciSioN.

Conclusion

Line 225 the authors state their laudable organisational aims. However it would be helpful if the manuscript could be expressed a little more objectively and scientifically.

Figure

The authors show the type and distribution of respondents. It would be helpful to know where the medical schools are distributed and how many medical students are present in each.

Reviewer opinion:

The authors have an original and interesting research concept. The authors are to be thanked for raising this debate and taking time to undertake this survey and prepare the manuscript.

I agree with them that this subject is of importance in the Universal health Coverage debate. They also clearly have great passion for their subject material. They have presented some interesting findings, and make some interesting proposals.

The big challenge here is the methodology. Details of this are noted in my comments. The likelihood of self-selection of respondents and the heterogeneity of the sample render the results impossible to analyse or to use for meaningful conclusions. The sample size is relatively small. Even with improved explanation of the methodology used, I think there would be too many concerns on methodology to enable the results to be published.

If the authors would want to have the study published in this journal, I would suggest redoing the questionnaire, and showing clearly how questions were asked. They could then administer it to a homogenous group such as all final year medical students in the country, and find a way to achieve a high response rate (preferably 80% or above).

Is it new? Yes

Is it of interest and value? Yes

Does the science stand up? No

Recommendation: not suitable for publication without complete revision of study (not just of manuscript).

6. PLOS authors have the option to publish the peer review history of their article (what does this mean?). If published, this will include your full peer review and any attached files.

Reviewer #1: No

Reviewer #2: No

---

## [Author Response · Author response to Decision Letter 0]

21 Feb 2020

Dear Editor and Dear reviewers,

We wish to thank you for taking the time to review our manuscript. The suggestions you made and the questions you raised were very helpful and accurate. We have addressed your comments and we feel this has substantially improved the quality of our manuscript. We look forward to reading your comments. 

Reviewers' comments:

Reviewer's Responses to Questions

Comments to the Author

5. Review Comments to the Author

Reviewer #1: The paper reads well but is primarily a discussion on global surgery. There is clearly a need to develop surgical services in Cameroon, but does the workforce need to advocate global surgery or to focus on developing surgical services in Cameroon itself? The latter is likely more important. There should be details included of the plans for Cameroon to develop their surgical services and whether the government is guided by global health targets.

Re: Dear reviewer, thank you for this valuable comment. The public health system in Cameroon is very centralized: key decisions are made at the Ministry of Health (MoH) level based on information gathered at the peripheral level (district hospitals). The information gathered peripherally is relayed to the central MoH by multiple mid-level administrative bodies. As a result, bureaucracy delays responses from the MoH and leads to a fragmentation of the public health system. 

Another problem is the earmarking of funds for diseases such as HIV/AIDS, Malaria and Tuberculosis. This has created a disparity between these high-profile diseases and other diseases. For example, most performance-based financing indicators are focused on infectious diseases, non-surgical maternal health and non-surgical child health. Consequently, the provision of care is skewed towards the incentivized diseases to the detriment of the other diseases. This gives the impression that the burden of the non-incentivized diseases is small or negligible. We believe the workforce can advocate for surgically amenable diseases if they advocate for the non-incentivized diseases. They have a role to play in surgical services delivery in Cameroon but at the moment their capacity is limited by non-recognition of the burden of surgically amenable diseases. 

The contribution of Mercy Ships is well made through the lead in training and providing a vital service for the local community.

I believe this paper is worthy of publication but more as a discussion of the need for countries (like Cameroon) to work towards achieving Global Surgery targets. The questionnaire is more of a side discussion.

Statistics are included in the paper but do not add to the analysis and should be removed.

Re: Dear reviewer, we share your opinion on the quality of the survey. Following advice from the academic editor, we re-administered the survey to a more homogenous population (final year medical students). We ran summary descriptive analysis and non-parametric bivariate analysis. As a result, we have rewritten the results and discussion sections to reflect this change.

Reviewer #2: I am interested that the lead researcher appears to be based in DR Congo and ethics was approved there, although the study was conducted in Cameroon. This seems to conflict with line 117 where the manuscript states that local ethical approval was obtained.

Re: Dear reviewer, thanks for taking the time to review our article. We found your comments accurate and beneficial. We had obtained IRBs at the academic institutions of the authors in Cameroon and DR Congo. We have attached the IRB obtained in Cameroon to this submission.

The introduction is nicely constructed and builds to the purpose of this study.

It would be very helpful to have some workforce data for Cameroon. Eg what percentage of doctors registered in Cameroon work in anaesthetic, obstetric or surgical disciplines? http://cm-minsante-drh.com/site/images/stories/Rapport_general_du_recensement01_12_2011_misenforme_FINAL05122001.pdf

Re: Thank you for your advice and kind words. We agree with you that having workforce data will make for a better read. We have included data on the surgical, obstetric and anesthesia workforce in the manuscript (lines 101-102).

Line 92 I am unsure what the authors mean when they say that global surgery in Cameroon is ‘led’ by InciSioN? Do they mean that InciSioN is one of the leading pressure groups promoting global surgery?

Re: Dear reviewer, we agree with you. We meant that InciSioN is one of the leading pressure groups promoting global surgery. We opted to exclude this statement as well as most mentions of InciSioN in the manuscript. As you rightfully noted our paper was monofocused.

Line 101-102: I feel that the authors could emphasize the wider potentials for the study. This study would not be appropriate for publication in PLOS ONE unless the application of the findings would be wider than Cameroon.

Re: Dear reviewer, we have rewritten our discussion to emphasize the wider potential of the study in the United States and low- and middle-income countries.

In Methods the authors first describe how they designed their survey line107. This section is brief and the reader only really knows which literature was read.

Re: Dear reviewer, thanks for your remark. Per your recommendation we have rewritten the methods and given additional details.

The questionnaire itself needs to be seen so that readers can see how questions were expressed, and could repeat the study in their own country or at a different time.

Re: We apologize for this mistake. We have added the questionnaire to the resubmission.

The authors then describe how they found their participants for the study. Their description is not very clear – either in terms of how they selected or accessed their target audience. They say that they used a convenience non-probability sampling method but do not reference this or explain how it worked, line 112.

They do not state how many people received the questionnaire, and hence the response rate is not evident.

It is very hard to know how representative their sample is likely to be.

Re: Dear reviewer, we have addressed recruitment, response rate and representativeness in our methods (lines 115-121).

In line 112,113 I am unsure why they chose medical Students and nurses and doctors including surgeons. It seemed to be anyone who was willing to complete the survey. The study would have more clarity if it was a single target audience. It is hard to imagine the respondents did not self-select as those interested in the topic. The groups sampled cross professions and training/trained demographics. For the medical students we are not told in which year of study they are currently. Clearly a first year medical student would have less chance of understanding global surgery than a final year student. We are not told if there is teaching on this subject in the medical student curriculum, and if so in which year of study it is taught?

The forms were anonymous but the authors say that they omitted members of the InciSioN group – how did they do this?

Re: Following your remarks and those of the academic editor we re-administered the survey to a homogenous population of seventh-year medical students. Our previous response rate had been affected by the conflicts in the North West and South West regions of Cameroon – where two of the six medical schools are located. We have been able to reach a much wider audience, ensuring generalizability.

Results

Line 121 there were only 45 respondents. We are not informed how many medical students are present in Cameroon, or indeed health care workers.

Re: Dear reviewer, we have re-administered the survey and rewritten both the methods and results to address this issue.

Line 129: do the authors mean most popular (well thought of) or popular (well known).

In the table:

Health professionals presumably means 18 doctors and one nurse?

Re: Dear reviewer, we have re-administered the survey to final year medical students and rewritten both the results and discussion.

The question ‘do you know about ‘global surgery’ appears confusing, and might be better expressed ‘have you previously heard of the term ‘global surgery’?’

Re: We agree with you esteemed Dr. The question is phrased as you suggested in our survey form which has been submitted along with this response.

For all the summarised section results it is unclear whether or not these are based on open or closed questions. For example were the organisations named and the respondents stated whether or not they had heard of them, or were they free text responses conceived by respondents?

Re: We have submitted the survey form along with this letter. The organizations were listed and the respondents could list an organization if it was not on the list.

Line 144 some results are given in the text but not in the table whilst others are in the table and repeated in the text. This needs to be consistent.

Re: Thanks Dr. We have attended to this in our new Table 1 (line 145).

Having not seen the questionnaire I am unsure whether any questions explored the place of trauma and injury in surgically treatable disease. This is particularly relevant to the perception of surgically treatable disease as a public health issue.

Re: We agree with you. We opted to limit our questions to the basic concepts of global surgery because there are other key concepts such as emergency obstetric care and emergency laparotomies that are just as deserving but would have lengthened the survey.

Discussion

The discussion is quite long but rather mono-focussed.

There is no comment on similar literature.

Re: Thanks for your insightful comment. We have rewritten the discussion.

There is some discussion regarding why respondents responded as they did.

Much of the focus seems to bear on proposed solutions to spread the word regarding global surgery, and a bit of publicity for InciSioN Cameroon. It would be helpful if the authors explored the issue of current knowledge and understanding of global surgery without such particular emphasis on InciSioN.

Re: We have rewritten the discussion to be less monofocused.

Conclusion

Line 225 the authors state their laudable organisational aims. However, it would be helpful if the manuscript could be expressed a little more objectively and scientifically.

Re: You are right dear reviewer. We have rewritten the conclusion to be less passionate and more scientific.

Figure

The authors show the type and distribution of respondents. It would be helpful to know where the medical schools are distributed and how many medical students are present in each.

Re: Following your advice, we have redesigned the figure to show the location of the medical schools.

Reviewer opinion:

The authors have an original and interesting research concept. The authors are to be thanked for raising this debate and taking time to undertake this survey and prepare the manuscript.

I agree with them that this subject is of importance in the Universal health Coverage debate. They also clearly have great passion for their subject material. They have presented some interesting findings, and make some interesting proposals.

The big challenge here is the methodology. Details of this are noted in my comments. The likelihood of self-selection of respondents and the heterogeneity of the sample render the results impossible to analyse or to use for meaningful conclusions. The sample size is relatively small. Even with improved explanation of the methodology used, I think there would be too many concerns on methodology to enable the results to be published.

If the authors would want to have the study published in this journal, I would suggest redoing the questionnaire, and showing clearly how questions were asked. They could then administer it to a homogenous group such as all final year medical students in the country, and find a way to achieve a high response rate (preferably 80% or above).

---

## [Decision Letter · Decision Letter 1]

23 Mar 2020

PONE-D-19-34339R1

Exploring the knowledge and attitudes of Cameroonian medical students towards global surgery: a web-based survey

PLOS ONE

Dear Dr. Kanmounye,

Thank you for submitting your manuscript to PLOS ONE. After careful consideration, we feel that it has merit but does not fully meet PLOS ONE’s publication criteria as it currently stands. Therefore, we invite you to submit a revised version of the manuscript that addresses the points raised during the review process.

We would appreciate receiving your revised manuscript by May 07 2020 11:59PM. To enhance the reproducibility of your results, we recommend that if applicable you deposit your laboratory protocols in protocols.io, where a protocol can be assigned its own identifier (DOI) such that it can be cited independently in the future. For instructions see: http://journals.plos.org/plosone/s/submission-guidelines#loc-laboratory-protocols

We look forward to receiving your revised manuscript.

Kind regards,

James G. Wright

Academic Editor

PLOS ONE

Reviewers' comments:

Reviewer's Responses to Questions

**Comments to the Author**

1. If the authors have adequately addressed your comments raised in a previous round of review and you feel that this manuscript is now acceptable for publication, you may indicate that here to bypass the “Comments to the Author” section, enter your conflict of interest statement in the “Confidential to Editor” section, and submit your "Accept" recommendation.

Reviewer #1: All comments have been addressed

Reviewer #2: All comments have been addressed

2. Is the manuscript technically sound, and do the data support the conclusions?

Reviewer #1: Yes

Reviewer #2: Yes

3. Has the statistical analysis been performed appropriately and rigorously? 

Reviewer #1: Yes

Reviewer #2: I Don't Know

4. Have the authors made all data underlying the findings in their manuscript fully available?

Reviewer #1: Yes

Reviewer #2: Yes

5. Is the manuscript presented in an intelligible fashion and written in standard English?

Reviewer #1: Yes

Reviewer #2: Yes

6. Review Comments to the Author

Reviewer #1: The authors should be congratulated for a thorough reworking of the paper and their commitment demonstrated by carrying a wider survey. I believe this paper is a useful contribution to the literature and I hope the aims of there authors to raise the profile of surgical care in Cameroon will be met.

2 minor comments:

Line 39: only medical students completed the questionnaire

Line 148: typo- remove 'were'

Reviewer #2: Exploring the knowledge and attitudes of Cameroonian medical students towards global surgery: a web based survey

Reviewer comments:

Answering questions raised: the authors have been at pains to answer the questions and respond to the issues raised in the review of the first manuscript. They have gone to the lengths of completely re-administering the questionnaire and rewriting the whole manuscript. Effectively they have gone to the length of starting from scratch with the same concept but delivered with a greatly enhanced methodology.

Abstract: in results I would say 58% of respondents were male – rather than ‘most’.

The authors say that the mean knowledge level was 47%. The terminology ‘knowledge level’ is difficult for the reader to understand or quantify. The terminology assuming ‘numbers of roles’ is not easily understood by the reader.

Introduction:

This is nicely constructed with quotations which highlight the importance of the topic. It builds towards the purpose of the study. There is a slightly abrupt transition from the global surgical need to the knowledge and opinions of medical students. Perhaps a bridging sentence such as ‘since Medical Students form the national potential in workforce capacity development, their attitudes towards global surgery are of paramount importance.’

Methodology:

This description is now succinct and much easier to understand than previously. The target group is homogenious and quantifiable.

A pilot study was used to hone the quality of the questionnaire.

I would still really like to see the survey rather than just a link to it. This would be one of my main recommendations.

Results:

Since the aim is to look at Medical student knowledge and attitude, I am unsure as to why the results are divided by sex? The aim was not to look as sex differences in attitude and knowledge. No gross differences emerge, so why not simply present the results for all respondents and say that no significant differences were observed based on the sex of the student?

I am not sure that figure 1 adds a huge amount. The main question seems to be - of the six medical schools in the country, what was the variation in participation rate? This may affect the generalizability of the findings.

The results summarized in the text lines 148-174 are useful and interesting.

Line 167 assuming numbers of roles is not understood to this reviewer and therefore probably not to most readers. Can this be defined?

Figure 2 is quite useful in allowing the reader to compare the strength of opinion relating to the responses with Likert grades.

Discussion:

The discussion is much improved from last time with some comparison to other studies and broader assessment of the impact of educating medical students regarding global surgery.

I am interested that the authors do not recommend a change in the undergraduate curriculum to include at least on lecture on global surgery, and perhaps another on universal health access? What they seem to have discovered is that social media and external events are moderately effective in filling a gap in the curriculum. It seems reasonable that they should advocate for that gap to be filled.

When discussing the limitations the authors focus on survey uptake and whether the sample is representative. In fact this has become a strength of the study with more than 80% response rate.

Other limitations would include:

Variations in understanding the questions; limitations of areas of questioning encompassed by the survey; unproven generalizability of these results to other lmics.

Conclusion:

The conclusion is a little wordy. The last sentence is a good one but perhaps should say ‘in global surgical education and activities’.

Overall reviewer opinion:

This is a vastly improved and totally reworked manuscript. It is an important area of research. It is now a coherent study. I would recommend it as being suitable for publication subject to a few minor edits.

7. PLOS authors have the option to publish the peer review history of their article (what does this mean?). If published, this will include your full peer review and any attached files.

Reviewer #1: Yes: Richard O.E. Gardner

Reviewer #2: No

---

## [Author Response · Author response to Decision Letter 1]

24 Mar 2020

Response to reviewers below the "R:"

Reviewer #1: The authors should be congratulated for a thorough reworking of the paper and their commitment demonstrated by carrying a wider survey. I believe this paper is a useful contribution to the literature and I hope the aims of there authors to raise the profile of surgical care in Cameroon will be met.

R: 

Dr Gardner, this was possible thanks to you. Your comments greatly improved our manuscript.

2 minor comments:

Line 39: only medical students completed the questionnaire

R: We have corrected the sentence. (Line 39-40)

Line 148: typo- remove 'were'

R: Thanks. The typo has been corrected. (Line 151)

Reviewer #2: Exploring the knowledge and attitudes of Cameroonian medical students towards global surgery: a web based survey

Reviewer comments:

Answering questions raised: the authors have been at pains to answer the questions and respond to the issues raised in the review of the first manuscript. They have gone to the lengths of completely re-administering the questionnaire and rewriting the whole manuscript. Effectively they have gone to the length of starting from scratch with the same concept but delivered with a greatly enhanced methodology.

R: 

Thank you for your comments and kind words. You made this process enjoyable and rewarding. We learnt a lot.

Abstract: in results I would say 58% of respondents were male – rather than ‘most’.

R:

Thank you for pointing this out. We have made the changes suggested. (Lines 50-51)

The authors say that the mean knowledge level was 47%. The terminology ‘knowledge level’ is difficult for the reader to understand or quantify. The terminology assuming ‘numbers of roles’ is not easily understood by the reader.

R:

We have reworded the text and we hope it is clearer for the audience. (Line 53-57)

Introduction:

This is nicely constructed with quotations which highlight the importance of the topic. It builds towards the purpose of the study. There is a slightly abrupt transition from the global surgical need to the knowledge and opinions of medical students. Perhaps a bridging sentence such as ‘since Medical Students form the national potential in workforce capacity development, their attitudes towards global surgery are of paramount importance.’

R:

Thank you. We have made changes to the introduction. (Lines 105-107)

Methodology:

This description is now succinct and much easier to understand than previously. The target group is homogenious and quantifiable.

A pilot study was used to hone the quality of the questionnaire.

I would still really like to see the survey rather than just a link to it. This would be one of my main recommendations.

R: 

We are sorry for the inconvenience caused. Following your initial comments, we had submitted the questionnaire as a supplemental file. However, it seems you did not get the file. We have resubmitted the questionnaire.

Results:

Since the aim is to look at Medical student knowledge and attitude, I am unsure as to why the results are divided by sex? The aim was not to look as sex differences in attitude and knowledge. No gross differences emerge, so why not simply present the results for all respondents and say that no significant differences were observed based on the sex of the student?

R:

Thank you for your comments. Although the sex disparities were not statistically significant, the differences between the two genders were important. It is possible that we were not powered to detect the differences in gender. Surgery and global surgery are fields dominated by men and we felt it was critical to report on the gender differences (Odds ratio of female/male 1.61 and 1.40). We presented these results to highlight this too. With your permission we wish to keep these results.

I am not sure that figure 1 adds a huge amount. The main question seems to be - of the six medical schools in the country, what was the variation in participation rate? This may affect the generalizability of the findings.

R:

Dear reviewer, we added figure 1 at the request of reviewer 1 who felt it was important to show the distribution of participants.

The results summarized in the text lines 148-174 are useful and interesting.

R:

Thank you 

Line 167 assuming numbers of roles is not understood to this reviewer and therefore probably not to most readers. Can this be defined?

R:

Dear reviewer, thank you for your comments. We have reworded the sentence. (Lines 170-172)

Figure 2 is quite useful in allowing the reader to compare the strength of opinion relating to the responses with Likert grades.

R:

Thank you.

Discussion:

The discussion is much improved from last time with some comparison to other studies and broader assessment of the impact of educating medical students regarding global surgery.

R:

We are honored by your comments and grateful for your help.

I am interested that the authors do not recommend a change in the undergraduate curriculum to include at least on lecture on global surgery, and perhaps another on universal health access? What they seem to have discovered is that social media and external events are moderately effective in filling a gap in the curriculum. It seems reasonable that they should advocate for that gap to be filled.

R:

Thank you for pointing this out. We have added your suggestion to the discussion. (Line 187-188)

When discussing the limitations the authors focus on survey uptake and whether the sample is representative. In fact this has become a strength of the study with more than 80% response rate.

Other limitations would include:

Variations in understanding the questions; limitations of areas of questioning encompassed by the survey; unproven generalizability of these results to other lmics. 

R:

We have added these to our limitations. (Lines 227-229)

Conclusion:

The conclusion is a little wordy. The last sentence is a good one but perhaps should say ‘in global surgical education and activities’.

R:

We have rewritten the conclusion per your recommendation. (Conclusion)

Overall reviewer opinion:

This is a vastly improved and totally reworked manuscript. It is an important area of research. It is now a coherent study. I would recommend it as being suitable for publication subject to a few minor edits.

R:

This would not have been possible without your help. Thank you for your patience and assistance all throughout this process.

---

## [Decision Letter · Decision Letter 2]

14 Apr 2020

Exploring the knowledge and attitudes of Cameroonian medical students towards global surgery: a web-based survey

PONE-D-19-34339R2

Dear Dr. Kanmounye,

We are pleased to inform you that your manuscript has been judged scientifically suitable for publication and will be formally accepted for publication once it complies with all outstanding technical requirements.

With kind regards,

James G. Wright

Academic Editor

PLOS ONE

Additional Editor Comments (optional):

Reviewers' comments:

Reviewer's Responses to Questions

**Comments to the Author**

1. If the authors have adequately addressed your comments raised in a previous round of review and you feel that this manuscript is now acceptable for publication, you may indicate that here to bypass the “Comments to the Author” section, enter your conflict of interest statement in the “Confidential to Editor” section, and submit your "Accept" recommendation.

Reviewer #2: All comments have been addressed

2. Is the manuscript technically sound, and do the data support the conclusions?

Reviewer #2: (No Response)

3. Has the statistical analysis been performed appropriately and rigorously? 

Reviewer #2: (No Response)

4. Have the authors made all data underlying the findings in their manuscript fully available?

Reviewer #2: (No Response)

5. Is the manuscript presented in an intelligible fashion and written in standard English?

Reviewer #2: (No Response)

6. Review Comments to the Author

Reviewer #2: (No Response)

7. PLOS authors have the option to publish the peer review history of their article (what does this mean?). If published, this will include your full peer review and any attached files.

Reviewer #2: No

---

## [Editor Report · Acceptance letter]

21 Apr 2020

PONE-D-19-34339R2 

Exploring the knowledge and attitudes of Cameroonian medical students towards global surgery: a web-based survey 

Dear Dr. Kanmounye:

I am pleased to inform you that your manuscript has been deemed suitable for publication in PLOS ONE. Congratulations! Your manuscript is now with our production department. 

With kind regards,

on behalf of

Professor James G. Wright 

Academic Editor

PLOS ONE